# Native Point Defect Measurement and Manipulation in ZnO Nanostructures

**DOI:** 10.3390/ma12142242

**Published:** 2019-07-12

**Authors:** Leonard Brillson, Jonathan Cox, Hantian Gao, Geoffrey Foster, William Ruane, Alexander Jarjour, Martin Allen, David Look, Holger von Wenckstern, Marius Grundmann

**Affiliations:** 1Department of Physics and Department of Electrical & Computer Engineering, The Ohio State University, Columbus, OH 43210, USA; 2Department of Physics, The Ohio State University, Columbus, OH 43210, USA; 3Department of Electrical and Computer Engineering, The Ohio State University, Columbus, OH 43210, USA; 4Department of Physics, Cornell University, Ithaca, NY 14853, USA; 5Department of Electrical and Computer Engineering, University of Canterbury, Christchurch 8140, New Zealand; 6Air Force Research Laboratory, Sensors Directorate, WPAFB, OH 45433, USA; 7Semiconductor Research Center, Wright State University, Dayton, OH 45435, USA; 8Institut für Experimentelle Physik II, Universität Leipzig, Linnéstr. 5, 04103 Leipzig, Germany

**Keywords:** native point defects, cathodoluminescence spectroscopy, nanowires, nanostructures, interface, electronic measurement

## Abstract

This review presents recent research advances in measuring native point defects in ZnO nanostructures, establishing how these defects affect nanoscale electronic properties, and developing new techniques to manipulate these defects to control nano- and micro- wire electronic properties. From spatially-resolved cathodoluminescence spectroscopy, we now know that electrically-active native point defects are present inside, as well as at the surfaces of, ZnO and other semiconductor nanostructures. These defects within nanowires and at their metal interfaces can dominate electrical contact properties, yet they are sensitive to manipulation by chemical interactions, energy beams, as well as applied electrical fields. Non-uniform defect distributions are common among semiconductors, and their effects are magnified in semiconductor nanostructures so that their electronic effects are significant. The ability to measure native point defects directly on a nanoscale and manipulate their spatial distributions by multiple techniques presents exciting possibilities for future ZnO nanoscale electronics.

## 1. Introduction

ZnO research has expanded considerably in the past few years as its outstanding optoelectric [1], microelectronic [2], and piezoelectric [3] properties have become apparent [4,5,6,7]. Indeed, ISI lists over 9000 ZnO-related research papers published in 2018 alone, professional journals and some of the largest symposia at scientific meetings have focused on ZnO, e.g., the Fall Materials Research Society, the Electronic Materials Conference, and the biennial International Workshop on ZnO and Related Materials, which just completed its 10th cycle. ZnO has fundamental properties that make it well suited for photonic and electronic applications such as UV light-emitting diodes [8], laser diodes [9] for white lighting, high-capacity DVDs, and transparent transistors [10] for displays.

ZnO nanostructures are now a rapidly growing component of ZnO papers overall, with over 5000 publications in just the past 5 years due to the relative ease of growth and numerous applications of the materials in microelectronics, photonics, and sensing [11,12]. The ease of nanostructure growth from ZnO precursors enables the fabrication of high surface-to-volume sensors and other surface sensitive devices for the detection of gases and biomolecules. Even lasing has been demonstrated in ZnO nanorods [13]. The biocompatibility of ZnO enables bio-implanted nanogenerators [14] and degenerately-doped ZnO enables plasmonic coupling of photons with electronics [4]. Recent studies have shown that native point defects are present in the “bulk” of ZnO nanowires and not just on their surfaces [15,16]. These defects inside nanowires and at their metal junctions can dominate the electrical contact properties, yet they can be moved by applied electric fields [17] or removed by ion beam techniques [18], Since defect segregation to free surfaces and interfaces is common among semiconductors [19], and its effect is magnified in semiconductor nanostructures, these results can have more general significance. In Section 2, we first show that native point defects can affect electronic properties. Section 3 shows how these defects distribute inside nanostructures, while Section 4 shows how these distributions alter electronic properties. Finally, Section 5 provides examples of manipulating native point defects to control electronic properties.

## 2. Electronic Properties of ZnO Nanostructures

The electronic properties of ZnO, as well as those of other semiconductor nanostructures, are relatively unexplored due to the challenges posed by their sensitivity to ambient atmosphere, illumination, strain, and physical size. Likewise, researchers have not considered the impact of native point defects on the electronic properties of these materials due to the difficulty in measuring them directly. As a semiconductor nanowire example in general, Figure 1 illustrates current-voltage measurements of a metal contact to a Ge nanowire. The conductance behavior of an end-on Au catalyst contact to the nanowire pictured in Figure 1a exhibits I–V transport that depends strongly on nanowire diameter (Figure 1b) [20]. Modeling the nanowire contact electrostatics, Léonard et al. showed that thermionic emission contributions to charge transfer across the junction are negligible compared with charge transfer associated with electron-hole recombination in the semiconductor depletion region. The conductance increase with decreasing wire diameter shown in Figure 1c is counterintuitive, yet it can be explained by the presence of recombination centers within the depletion region which increase with decreasing wire diameter. The ideality factors corresponding to this increase in recombination-dominated transport also increase (Figure 1c inset) due to gap states within the depletion region. Since the Ge nanowires are of high purity, native point defects, rather than impurities, must account for these gap states, although they are not directly observed.

Electrical measurements of ZnO nanowire device structures exhibit dramatic sensitivity to ambient conditions and applied fields. Weissenberger et al. used a field effect transistor (FET) structure based on a ZnO nanowire to demonstrate the strong hysteresis effects of wire conductivity and transistor threshold voltage with applied bias [21]. Figure 2a illustrates a source-drain current (I_ds_) threshold voltage dependence on applied gate voltage V_G_ with constant source-drain voltage (V_ds_). Similarly, Figure 2b–d show a strong dependence on applied bias, direction, and sweep rate, which were interpreted as electric field-induced oxygen desorption and adsorption under ambient conditions. Lower voltage measurements under high vacuum conditions (10^−5^ mbar) in (d) alter the hysteresis in (c) significantly, but still exhibit measurable hysteresis. Besides illustrating the importance of ambient effects, these measurements also show how measurement procedures can influence the apparent electronic properties of ZnO nanowires. Such bias effects on nanowire electronic properties are not unique. They include, for example, the effect of bias voltage on the adsorption of specific molecules [22] and the voltage dependence of Schottky barrier distribution parameters and peak transistor gain [23]. Schlenker et al. have pointed out how defect density and surface morphology can affect resistivity [24]. In general, such electronic measurements have not taken into account the presence and distribution of native point defects inside ZnO nanostructures and their electrical activity.

## 3. Defect Distributions in ZnO

Defects can strongly affect the electronic behavior of semiconductor nanostructures, depending on their physical nature, density, and spatial distribution. While the physical nature and electrical activity of native point defects in ZnO crystals has been studied extensively [25], defects in ZnO nanostructures have been relatively unexplored. Until recently, researchers had considered the behavior of defects only at the surfaces of nanostructures. For example, Figure 3 illustrates how electronic conduction through nanowires changes with above band gap excitation and oxygen adsorption [26]. Initially, above band gap excitation of the nanowire in Figure 3a increases free electron and hole densities, increasing collected current in the circuit. Adsorbed oxygen molecules in the dark in Figure 3b trap negative charge at surface states, increasing band bending and depletion widths to reduce nanowire conduction. Above band gap illumination in Figure 3c creates additional electron-hole pairs that recombine with trapped surface charge, desorbing O_2_ molecules, reducing band bending, and increasing electrical conduction.

Scanning electron microscopy (SEM) and transmission electron microscopy (TEM) studies have now shown that defects are also present inside semiconductor nanostructures. Thus, extended defects such as dislocations and stacking faults are observable and correlated to structural features using spatial imaging and cathodoluminescence spectroscopy (CLS) on a nanoscale [27,28,29,30,31]. Electron microscope studies have described how ZnO properties vary within interfaces, micro- and nano- structures and the effect of growth and processing methods [32,33]. For example, Figure 4 illustrates CL spectra of n-ZnO nanorod/GaN heterostructures grown in a solution of zinc (II) acetate dehydrate (Zn(Ac)) and KOH in methanol [33]. Measured in cross section, CLS of the ZnO nanorods near the GaN base (blue dot and blue curves) exhibit multiple deep level emissions with maxima extending across 1.85–2.2 eV, depending on the molarity. Figure 4 shows that intensities of these defects relative to the NBE emission vary with spatial location within individual nanowires and their proximity to the GaN substrate. Furthermore, the relative intensities of these defects increase according to the molarity of the growth solution, showing their high sensitivity to specific concentrations of reactants during chemical synthesis. Based on the range of defect emission energies, the Zn(Ac):KOH ratio appears to influence the density and size of zinc vacancy V_Zn_ clusters [34]. Overall, multiple reviews of CLS applied to ZnO nanostructures are now available [35,36,37,38,39,40,41], and numerous studies have shown impurities and defects localized inside nanostructures [42,43,44,45,46].

Hyperspectral imaging (HSI) provides the ability to image the spatial distribution of defects inside nanostructures. With this capability, researchers can observe how defects in nanostructures localize in nanoscale spatial distributions, and how they can move under applied electric fields. Specific defects exhibit pronounced segregation toward the free surfaces of ZnO nanostructures. Figure 5 illustrates defect segregation toward the surfaces of ZnO nanorods [48] and tetrapods [41]. Figure 5a shows an SEI of ZnO nanorods. Maps of the NBE (Figure 5b) and green luminescence (GL) (Figure 5c) spatial distributions of peak integrated peak areas correspond to the Figure 5d CL spectrum. The inset represents the anti-correlation of GL and NBE emissions schematically. Figure 5e shows an SEI of 300–500 nm diameter ZnO tetrapods and the corresponding SEI map (f) of 2.5 eV integrated area, which extends more than 50 nm into the “bulk” and are attributed to oxygen vacancy-related (V_O_-R) defects.

Depth-resolved cathodoluminescence spectroscopy (DRCLS) measurements of individual nano- and micro- wires provide further evidence of defect distributions inside these wires. Figure 6 shows how the DRCLS beam generates depth profiles both across the diameter and into the bulk of an individual wire. Figure 6a illustrates how the electron beam excites luminescence across the diameter of a ZnO wire, while Figure 6b indicates the parameters used to simulate the optical emission taking into account the nanowire diameter, the excitation depth, and total internal reflection of the light inside the wire [15]. Figure 6c shows the profile of normalized defect intensity versus depth as a function of incident beam energy E_B_, while Figure 6d plots this normalized defect intensity (black dots) versus position across the wire diameter. The defect profile simulation generates profiles (blue, green, and red lines and dots) that approximate the actual data depending on a characteristic thickness *w* of defects toward the surface. Figure 6c shows an excellent fit between the energy-dependent depth profile into the wire and the *w* = 37 nm best fit of the 5 keV lateral profile. Similar to Figure 5, Figure 6 demonstrates that native point defects are present deep within the nano- and micro- wire solids.

The pronounced segregation from the ZnO nano-/microwire bulk toward the surface can be understood in terms of piezoelectric fields. The spontaneous polarization and piezoelectric constants for ZnO are relatively large and opposite in sign along versus transverse to the *c*-axis [49]. Figure 7 displays pronounced segregation extending over ~30 nm for ZnO [000$\overset{-}{1}$] V_O_-related [50] and [11$\overset{-}{2}$0] V_Zn_-related [51,52,53] emission intensities. The O-polar surface illustrated in Figure 7a generates a negative polarization, which attracts positively charged, ionized V_O_ defects. Conversely, the opposite polarization along [11$\overset{-}{2}$0] in Figure 7b attracts V_Zn_ defects toward the free surface. Note the lower V_O_ segregation in Figure 7b, which suggests a smaller electric field of opposite sign (perhaps due to n-type band bending). Thus, electric fields in bulk ZnO appear to affect defect distributions.

## 4. Impact of ZnO Defect Distributions on Electronic Measurements

The nature and density of native point defects in ZnO can have major effects on the electronic properties of metal-ZnO nanowire interfaces. As an example, Pt electrodes deposited on a single ZnO nanowire produce dramatically different defect emissions and electronic behavior that depend on the tapered nanowire diameter from the contact layout in Figure 8d. Figure 8a–c show DRCL spectra of three Pt-ZnO nanowire interfaces with wire diameters of 900, 600, and 400 nm respectively. Figure 8a shows deep level emission at Contact 1 centered at 2.35 eV corresponding to Cu_Zn_ sites that dominates NBE emissions along with lower intensity 1.61 eV emission corresponding to isolated V_Zn_ [34]. The Figure 8a insert shows ohmic I-V measurements between Contact 1 and 3, indicating that Contact 1 is ohmic. I–V characteristics in Figure 8b insert appear Schottky-like and DRCL spectra show only weak 1.59 eV emission again due to isolated V_Zn_. Depth profiles of Contacts 1 and 4 show Cu_Zn_, V_Zn_, as well as 2.45 eV V_O_ segregation that is much stronger for Contact 1. I–V characteristics between Contacts 1 and 5 display blocking character, i.e., no current in either direction. DRCL spectra show defect emissions that no longer dominate NBE emissions. Consistent with Figure 8, DRCL spectra along the wire length reveal monotonically decreasing defect intensities with decreasing wire diameter [18].

The defect intensities of this wire calibrated with Hall Effect measurements [54] show that defect intensities at Contact 1 are high enough to enable defect-assisted hopping conduction, whereas the lower densities at Contact 4 permit a Schottky barrier to form [18]. The narrow diameter Contact 5 coupled with n-type carrier compensation by V_Zn_ acceptors increases the depletion width radially to pinch off free carrier transport under the contact. Figure 9 illustrates schematically the radial variations of defect density in wire cross section, decreasing from Contact 1 to Contact 4 to Contact 5. Alongside each wire, energy band diagrams corresponding to each metal-wire interface indicate electron tunneling, rectification, and blocking, respectively [18].

## 5. Manipulating Native Point Defects and Controlling Electronic Properties

The previous section showed that the nature, density, and spatial distribution of native point defects can strongly impact electronic properties of nanostructures. In order to control these properties, several methods are now available to manipulate native point defects in these structures. These include: (i) removing defects by plasma treatments, (ii) removing defects by physically removing semiconductor volumes in which they reside, (iii) passivating defects to change their physical nature and electronic activity, and (iv) moving defects with applied electric fields. 

Remote oxygen plasma (ROP) treatments of oxide semiconductors such as ZnO [50], SrTiO_3_ [55], and Ga_2_O_3_ [56] have proved effective in removing V_O_-R defects by supplying activated but low kinetic energy oxygen that fills in the oxygen lattice vacancies without creating new defects. The ability to remove defects from near-surface lattice sites is also useful to identify the nature of defects. For example, in the case of Ga_2_O_3_, ROP treatment reduces the DRCLS signal of only one of a number of defect emissions, thereby identifying its V_O_-R nature [56]. Such treatments result in systematic and controllable defect reductions that are highest at the free semiconductor surface, decreasing into the bulk on a scale of tens of nm or more, depending on the ROP duration.

Low energy, focused ion beams can ablate away regions of high native defect density. Figure 6, Figure 7, and Figure 9 show that native point defects can segregate from semiconductor interiors toward the free surface. For nanowires such as those represented in Figure 9, a focused ion beam (FIB) of Ga with a 5 keV kinetic energy can ablate away the wire diameter. Figure 10a shows an SEI of a 700 nm diameter ZnO nanowire with a region reduced to 400 nm as the blue arrow indicates [18]. Stopping and Range of Ions in Matter (SRIM) simulations indicate damage limited to only 4.4 nm for 5 keV ion energies versus 14.7 nm for a typical ion implantation energy of 30 keV.

The removal of the outer 150 nm from the initial 350 nm wire radius reduces the density of defects near the metal-ZnO interface. Comparison of I–V characteristics for conduction through the milled ZnO-metal contact shows dramatic differences compared with an unmilled ZnO-metal contact. Figure 11a,b both exhibit Schottky-like rectification for the milled contact versus ohmic behavior for the unmilled contact. The 30 keV Ga-implanted counterelectrode in both cases is ohmic due to high Ga doping and lattice defects. The unmilled contact exhibits weakly ohmic behavior due to high defect density and tunneling so that milled to unmilled contact pairs exhibit Schottky-like behavior albeit less pronounced. In all three contact pairs, Figure 11b shows that forward current is reduced by 10^2^x, which is consistent with conventional current density J_0_ = A**T^2^ exp (−qΦ_SB_/kT) for Schottky barrier height qΦ_SB_. Reverse current at T = 80 K is reduced by more than 10^4^x, which is consistent with thermally-activated hopping conduction through the ZnO depletion layer. These results demonstrate that defect manipulation—in this case, defect removal—can produce Schottky barriers with the same metal (Pt) on the same nanowire that exhibits ohmic behavior.

An example of passivating defects to change their physical nature and electronic activity is the filling of oxygen vacancies in ZnO with hydrogen. As shown by Anderson and Janotti [57], hydrogen replacing oxygen at an otherwise oxygen vacancy results in the formation of a shallow n-type donor in ZnO. The resulting complex consists of a substitutional H_O_ and the four nearest-neighbor zinc atoms. This hydrogen passivation converts the V_O_ donor deep within the ZnO band gap into a shallow donor near the conduction band, thereby accounting for the increased free carrier density associated with the anion vacancy. In contrast, hydrogen passivation of Mg dopants accounts for the early difficulty of achieving p-type conductivity in GaN [58]. Acceptor activation and hole conduction result from high temperature annealing to remove the H bonding to Mg. Other impurities in ZnO nanostructures can inhibit particular defects. Thus, cobalt impurities appear to suppress paramagnetic native core-defects (CD) in ZnO nanorods, although the physical nature of the CD is not yet known [59].

DRCLS is now able to show that electric field biasing can move native point defects in ZnO bulk single crystals and nanowires. Figure 12a shows an SEI of the wire bonding layout used to apply voltage across two Pt contacts spaced 5 μm apart on a 3 μm diameter ZnO nanowire. Figure 12b shows hyperspectral maps of defect intensity area integrated over a spectral range of 2.3–2.6 eV and normalized by NBE area integrated over 3.2–3.3 eV. [60] Based on the electrode separation, an applied voltage V_A_ = 0.5 V corresponds to a voltage gradient of 1 kV/cm. Before applying voltage, defects appear uniformly distributed between electrodes. Peak emissions (not shown) at 2.35 eV and 2.0 eV dominate the DRCL spectra and correspond to Cu_Zn_ antisites and V_Zn_, respectively, both of which are acceptors. With increasing voltage up to V_A_ = 100 V, these defects appear to move toward the lower electrode. This movement becomes apparent for V_A_ of only 2–8 V. Likewise, wire resistivity gradually decreases by 25%, corresponding to an increase in free electron density. At V_A_ = 150 V, electrical breakdown occurs, causing a resistivity drop of more than 3 orders of magnitude. All electrical and optical measurements were performed in ultrahigh vacuum (UHV). UHV conditions minimize carbon contamination due to electron beam decomposition of CO_2_ onto the nanowire common to higher background pressure in conventional electron microscopes. In addition, UHV electrical measurements avoid complications due to adsorbed OH molecules that form conductive surface layers in air [50,61,62] as well as possible H_2_O molecular adsorption or desorption on ZnO induced by electrical bias [21]. Figure 12 also shows that the applied electric fields create non-uniform defect distributions—an accumulation of defects under the lower electrode as well as defect “tracks” both at the surface and near the wire center. Depending on the defect densities in these regions, defect-assisted tunneling can occur, creating conductive pathways. At very high densities, such defects can lead to dielectric breakdown.

The movement of defects under applied electric fields is significant for several reasons. Electrical measurements of nanowires typically employ applied voltages of 1–10 V or more; see Figure 2, for example. Since ZnO bulk crystals, as well as nanowires, commonly exhibit defect emissions, such applied voltages can produce similar defect movements. In turn, the electrical activity of such defects can produce substantial changes in free carrier density, depending on the defect density and its spatial distribution. Thus, the movement of acceptors driven by applied electric fields in Figure 12 serves to increase the free electron density intrinsic to the ZnO lattice, thereby reducing wire resistivity as observed. At high enough densities, defects accumulated near metal contacts can lower Schottky barrier heights [17] or lead to trap-assisted tunneling [18]. Finally, since band bending regions at Schottky barriers of <1 eV across depletion widths of <1 μm can produce built-in electric fields exceeding those discussed above, such contacts could induce defect movements as observed for Zn [18], SrTiO_3_ [63], and other semiconductors. Such mechanisms may contribute to the electric field effects described in Section 2.

## 6. Conclusions

The ability to measure and manipulate native point defects in ZnO opens new avenues to control their electronic properties. In general, techniques to remove native point defects from ZnO improve electronic properties by reducing electronic states in the band gap that serve to trap charge, induce photo-induced electron-hole pair recombination, and scatter free carriers. Removing electronically-active states that act as donors near interfaces can reduce tunneling through otherwise narrowed depletion regions of metal-semiconductor interfaces. Conversely, the addition of defects that act as acceptors in n-type ZnO can widen depletion regions and increase effective barrier heights. The density of such additional compensating centers must be balanced against the formation of trap-assisted tunneling pathways at high defect densities. Remote plasma treatments have already proved effective in removing defects inside ZnO over thicknesses more than 100 nm. Such treatments may well be effective for removing defects in nanowires of comparable dimensions. Low energy ion beam ablation may remove lattice volumes with high segregated defect densities. Applied electric fields to move native point defects may prove useful in creating such segregation, as well as adding or removing defects near surfaces and interfaces. Indeed, electric field-driven defect movement may provide the means to “refine” ZnO by extracting them from inside active device regions, then removing them altogether with plasma and/or ablation methods. In all these approaches, DRCLS provides the ability to measure the nature, density, and distribution of specific defects on a near-nanometer scale and in three dimensions. This combination of defect measurement and manipulation presents exciting possibilities for future ZnO nanoscale electronics.

## Figures and Tables

**Figure 1 materials-12-02242-f001:**
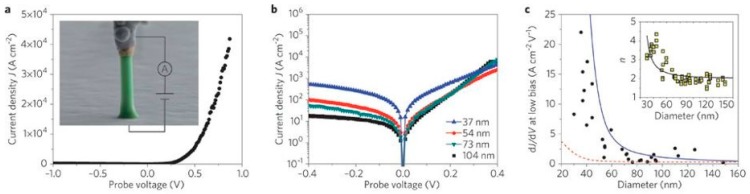
(**a**) End-on Au/Ge nanowire Schottky diode. (**b**) Current-voltage measurements for contacts to nanowires of different diameter. (**c**) Calculated conductance dI/dV based on a diameter-dependent (solid line) versus diameter-independent (dashed line) recombination time. Inset shows corresponding ideality factor versus wire diameter. Reprinted with permission from American Physical Society, Ridge, NY, USA [20].

**Figure 2 materials-12-02242-f002:**
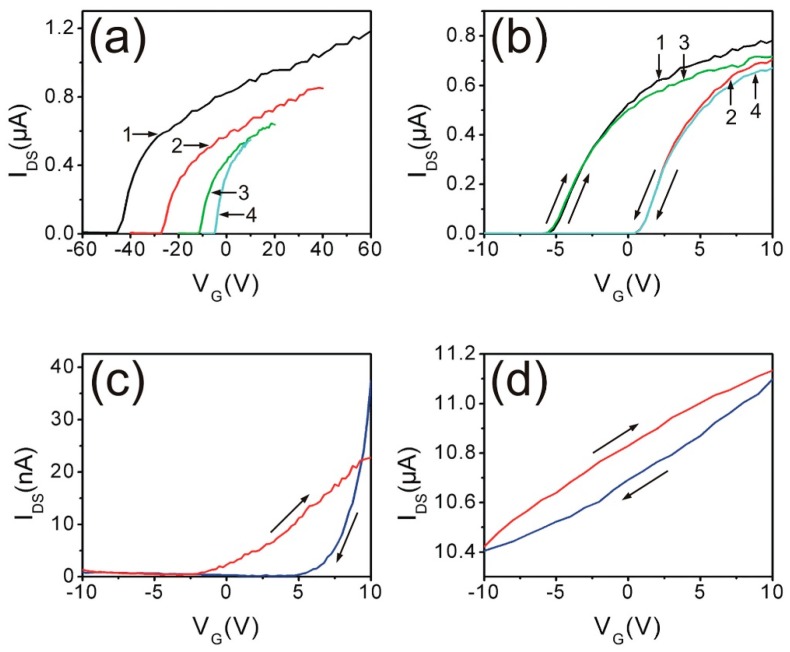
Gate voltage (V_G_)-dependent, 1 V source-drain bias, source-drain current (I_ds_(V_G_) of ZnO nanowire FET under ambient conditions for I_ds_(V_G_) sweep ranges (**a**) ±60 V, ±40 V, ±20 V, and ±10 V for sweeps labelled 1–4, respectively. (**b**) sweep range ±10 V for a doped and (**c**) undoped nanowire, and (**d**) the latter in high vacuum. Reprinted with permission from American Institute of Physics [21].

**Figure 3 materials-12-02242-f003:**
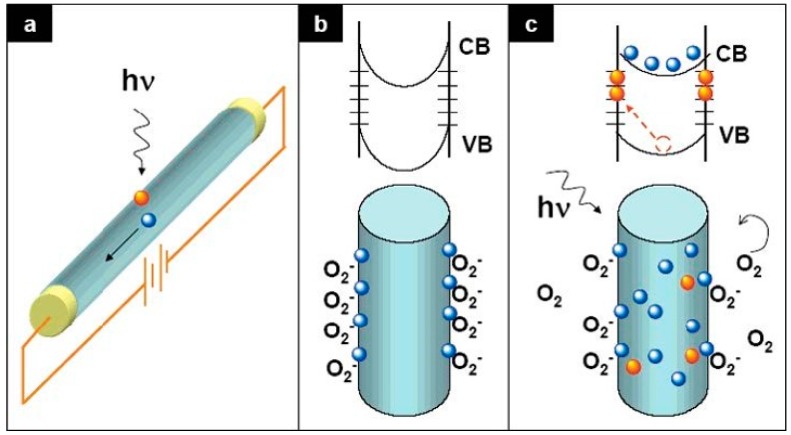
Influence of surface trap states on oxygen molecule adsorption, charge transfer, and band bending within a nanowire. (**a**) Photoexcitation increases nanowire free carrier density and conduction, which are reduced (**b**) by oxygen molecule adsorption, increased band bending, and reduced carrier density inside the wire. (**c**) Above band gap excitation of the O_2_—adsorbed wire reduces band bending as additional free carriers recombine with trapped surface charge and O_2_ molecules desorb. With permission from [26]. Copyright 2007 American Chemical Society.

**Figure 4 materials-12-02242-f004:**
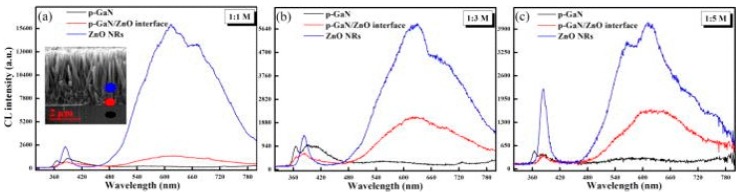
CL spectra of ZnO nanorod junctions with p-type GaN substrates measured in cross-section. (**a**) Inset shows a characteristic cross-sectional SEI of the heterostructure with blue, red, and black (from top to bottom) spots indicating spatial locations within the heterostructure corresponding to the blue, red, and black (from top to bottom) spectra in each of the figures. The 625–670 nm (1.85–2 eV) defect intensities relative to the 375 nm (3.3 eV) NBE peak emission increase from (**a**) 1:1 M to (**b**) 1:3 M to (**c**) 1:5 M. With permission from American Institute of Physics [47].

**Figure 5 materials-12-02242-f005:**
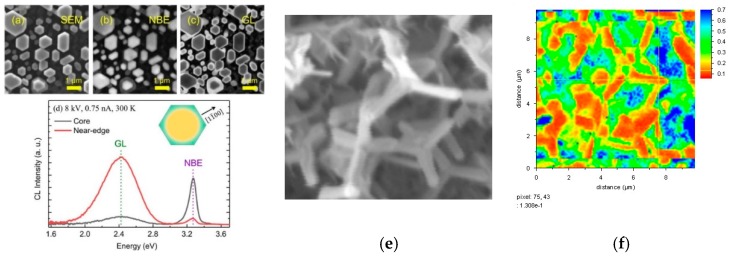
ZnO nanorod (**a**) SEI, region-of-interest (**b**) 3.32 eV NBE and (**c**) 2.46 eV GL intensity maps corresponding to the core and near-edge spectral features in (**d**). SEI of ZnO tetrapods (**e**) and corresponding HSI map (**f**) of NBE-normalized GL intensity extending 50–100 nm into the interior. With permission of John Wiley & Sons (**a**–**d**) [48] and Springer Nature (**e**,**f**) [41].

**Figure 6 materials-12-02242-f006:**
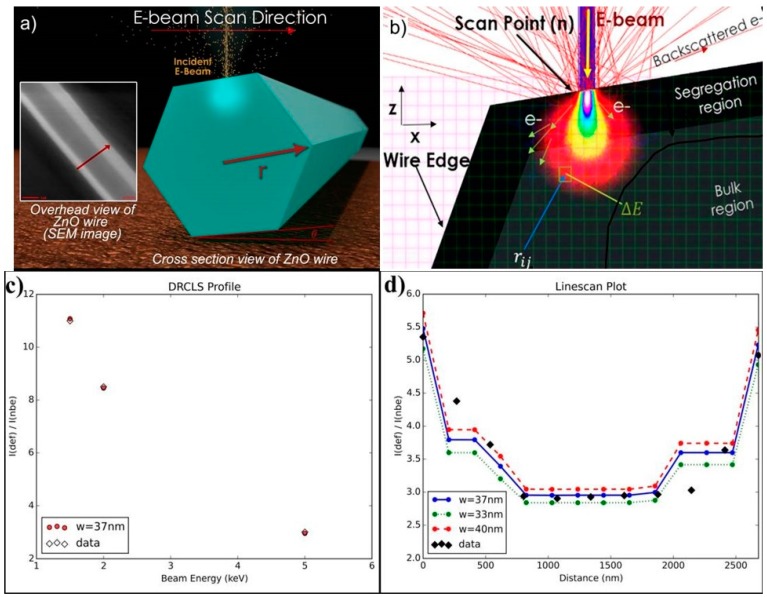
(**a**) Illustration of the line-scan CLS measurement process and SEI picture of a typical electron beam track across the ZnO wire diameter. (**b**) Illustration of electron-hole pair creation geometry and geometric parameters used for the defect profile simulation. (**c**) Single spot depth dependence measurement of I(defect)/I(NBE) intensity ratio versus incident beam energy E_B_. (**d**) 5 keV line-scan data (black dots) and best fit (blue line and dots) showing defect emissions extending nearly 1000 nm into the microwire bulk. Reprinted with permission from the Royal Society of Chemistry [15].

**Figure 7 materials-12-02242-f007:**
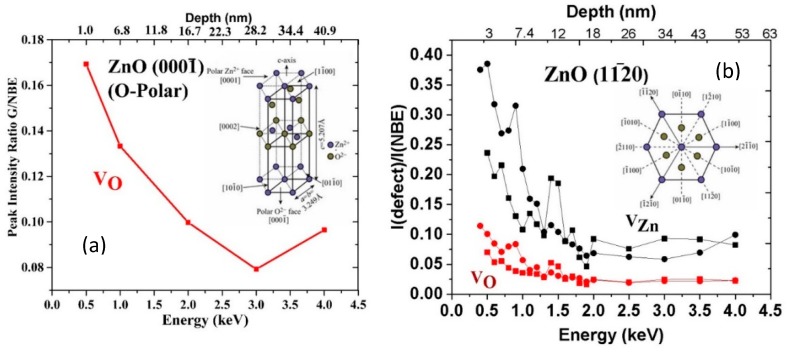
Defect segregation below a ZnO single crystal free surface versus crystal orientation. (**a**) [000$\overset{-}{1}$] V_O_ (2.5 eV) and (**b**) [11$\overset{-}{2}$0] V_Zn_ (1.7–2.1 eV) and V_O_ (2.4–2.5 eV) segregation. Higher V_O_ segregation in O- vs. Zn-polar ZnO also shown in ([52]) Reprinted with permission by American Institute of Physics Refs. ([51,52,53], respectively).

**Figure 8 materials-12-02242-f008:**
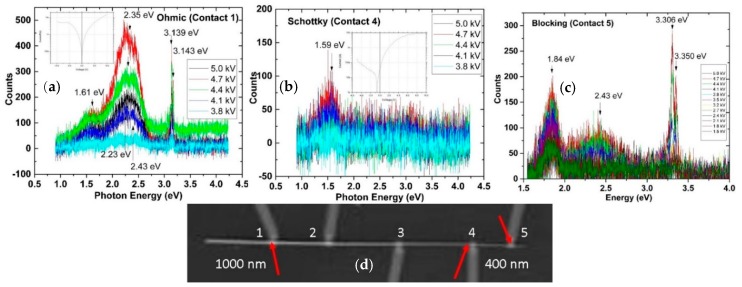
DRCLS of (**a**) ohmic, (**b**) Schottky, and (**c**) blocking contacts on the same ZnO nanowire (**d**) at locations indicated by arrows. Insets in (**a**,**b**) show ohmic and Schottky I–V characteristics, respectively. Deep level emissions at each contact depend strongly on the wire diameter.

**Figure 9 materials-12-02242-f009:**
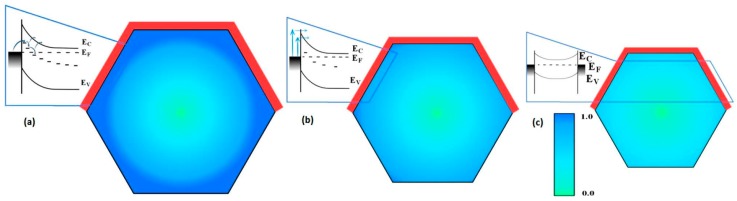
Schematic diagrams of band bending at Pt-ZnO nano/microwire contact for (**a**) 900 nm, (**b**) 600 nm, and (**c**) 400 nm diameter wires linked to the interfaces of their corresponding wires. Darker shading signifies higher acceptor density and defect-assisted hopping with increasing radius. With decreasing diameter, interface acceptor density decreases and contact behavior changes from transport by (**a**) trap-assisted tunneling to (**b**) Schottky rectification to (**c**) blocking regions extending radially from multiple faceted surfaces can almost fully deplete the 400 nm diameter nanowire. With permission from [18]. Copyright 2018 American Chemical Society.

**Figure 10 materials-12-02242-f010:**
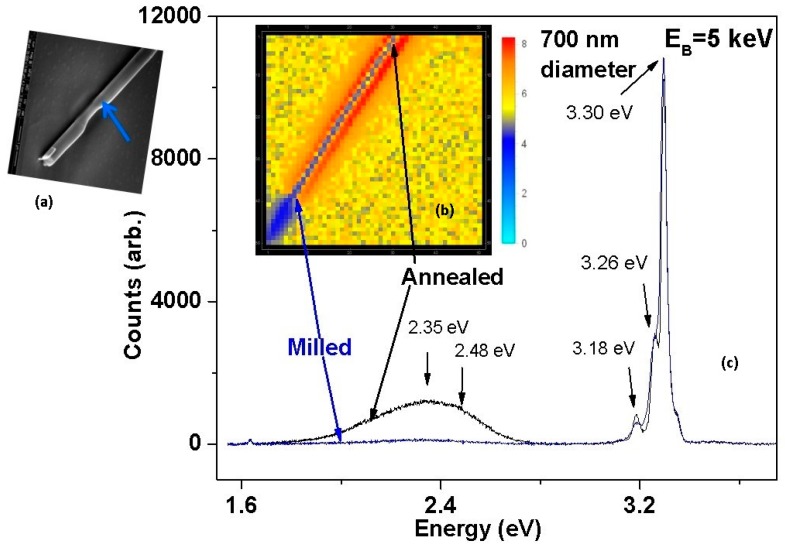
(**a**) SEI, (**b**) HSI, and (**c**) DRCL spectra obtained at E_B_ = 5 keV of a 700 nm diameter ZnO nanowire on SiO_2_ with a wire section milled down to 400 nm to remove segregated defects (lower left) versus a section e-beam annealed to promote additional defects (upper right). With permission from [18]. Copyright 2018 American Chemical Society.

**Figure 11 materials-12-02242-f011:**
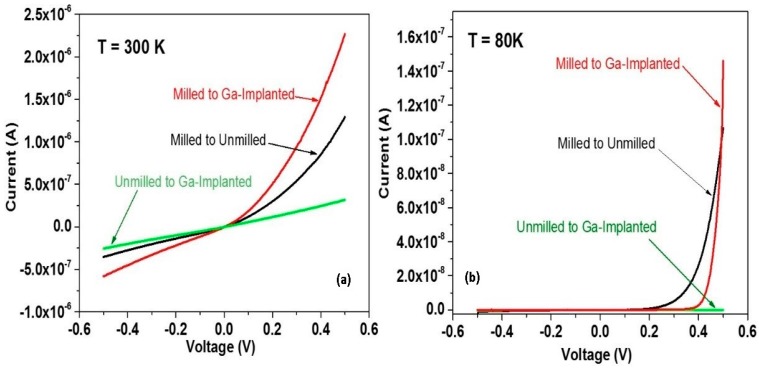
I–V characteristics for pairs of ZnO contacts at (**a**) 300 K and (**b**) 80 K. For both temperatures, the milled contact exhibits Schottky rectification versus ohmic conduction of the unmilled contact. With permission from [18]. Copyright 2018 American Chemical Society.

**Figure 12 materials-12-02242-f012:**
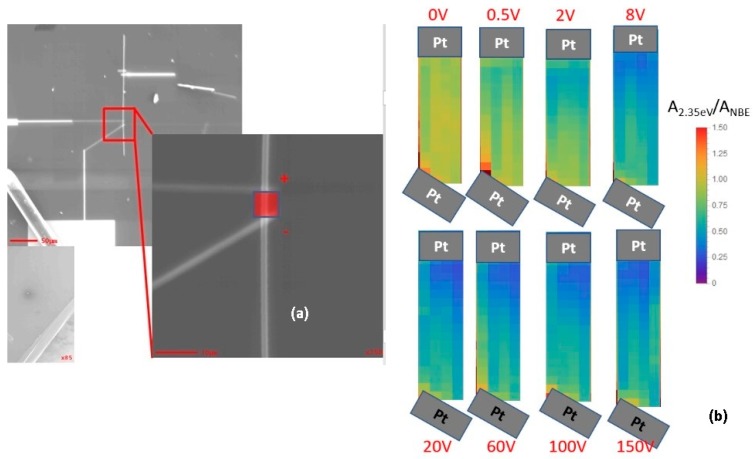
(**a**) Pt wire layout to apply voltage across two Pt contacts spaced 5 μm apart on a 3 μm diameter ZnO nanowire. (**b**) HSI maps of normalized defect intensity between Pt electrodes showing segregation of defects toward bottom electrode increasing with increasing applied voltage [60].

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
