# Peer review of "Native Point Defect Measurement and Manipulation in ZnO Nanostructures"

_materials, 2019, doi:10.3390/ma12142242_

Round 1

Reviewer 1 Report

The authors review on a topic which is essential in the development and future use of ZnO nanostructures in electronic devices. The information is clear and the manuscript gives a good sight on the state of the art of the influence of native defects in the electrical and electronic properties of ZnO nanostructures. However, a couple of questions could be addressed to improve the clarity of the results shown:

- In page 8, line 3, the authors use DRCLS acronym which has not been previously defined. I guess it is Deep Resolved Cathodoluminescence Spectra, but it has to be mentioned in the text.

- In the same page, figure 7.  In part a) the peak intensity ratio of oxygen vacancies-related band is shown. Are the oxygen vacancies the only contribution to defect emission in the samples investigated?  What energy or wavelength is monitored to study this peak intensity?. I have essentially the same questions for part b), what energy peak has been selected to calculate the ratio shown? Regarding the ZnO surface in part b), is it a polar surface? Are there any similar results in the Zn-polar surface (0001)? It would be interesting to compare results in both polar surfaces.

After these questions are adressed, the manuscript could be published.

Author Response

We thank Reviewer 1 for his/her favorable review and constructive comments.

 On page 8, line 3, we have now added “Depth-resolved cathodoluminescence spectroscopy (DRCLS) to this sentence.

On page 7 in part a), oxygen vacancies are the only evident contribution to defect emission in the samples investigated although the 2.5 eV peak emission is relatively broad.

On page 7 in part b), the normalized area between 1.7 and 2.1 monitors the zinc vacancies. The [112-0] direction is non-polar but with opposite polarization from the [000-1] polar face as already mentioned on page 8. Higher VO segregation in O- vs. Zn-polar ZnO also appears in [52(b)] with a second reference (to avoid renumbering).

Accordingly the Figure 7 caption now reads: “Figure 7. Defect segregation below a ZnO single crystal free surface versus crystal orientation.(a) [000ī] O-polar VO (2.5 eV) and (b) [112-0] VZn (1.7-2.1 eV) and VO (2.4 – 2.5 eV) segregation. Higher VO segregation in O- vs. Zn-polar ZnO also shown in [52(b)]. Reprinted with permission by American Institute of Physics Refs. [51,52, respectively]”, where reference 52(b) is “Y. Dong, Z-Q. Fang, D.C. Look, G. Cantwell, J. Zhang, J.J. Song, and L.J. Brillson, “Zn- and O-face effects at ZnO surfaces and metal interfaces,” Appl. Phys. Lett. 93, 072111 (2008). doi: 10.1063/1.2974983”

Reviewer 2 Report

This work overviews the techniques to measure and manipulate native point defects in ZnO nanostructures, which is important in order to precisely control their electronic properties. The review comments on the recent findings for manipulation of defects both in the bulk and in the interface of ZnO nanostructures, indicating how the effective future ZnO nanoscale electronics could be built.

There are still some points, which I would suggest for the authors to consider:

Authors start the explanation using Ge nanowires as and example without much explanation. Since the review is dedicated exclusively to ZnO nanostructures, Ge examples can be misleading. Figure 1 and explanation of I-V transport are referred to Ge nanowires, without much explanation how it is connected to the main topic of the review. The best would be to use ZnO nanostructure data at least for the figure 1, or elaborate more on why Ge was used as an example, and how Ge nanowire measurements are similar/different from ZnO. This would be more clear for the reader, who is not has no connection to Ge nanomaterial field.

figure 8 caption " Insets in (a) and (b) show ohmic and blocking I-V characteristics, respectively", however (b) shows Schottky.

Author Response

We thank Reviewer 2 for his/her favorable review and constructive comments.

In order to address any confusion concerning the Ge example shown in Figure 1, Page 2, Section II, lines 5 and 6 now read: “As a semiconductor nanowire example in general, Figure 1 illustrates current-voltage measurements of a metal contact to a Ge nanowire.”

In the Figure 8 caption, “blocking” is replaced by “Schottky” so that the caption now reads: “Figure 8. DRCLS of (a) ohmic, (b) Schottky, and (c) blocking contacts on the same ZnO nanowire (d) at locations indicated by arrows. Insets in (a) and (b) show ohmic and Schottky I-V characteristics, respectively. Deep level emissions at each contact depend strongly on the wire diameter.”